# Views of commissioners, managers and healthcare professionals on the NHS Health Check programme: a systematic review

Katie Mills,[1] Emma Harte,[2] Adam Martin,[3] Calum MacLure,[2] Simon J Griffin,[1,4] Jonathan Mant,[1] Catherine Meads,[5] Catherine L Saunders,[1] Fiona M Walter,[1] Juliet A Usher-Smith[1]

[1]The Primary Care Unit, Institute of Public Health, University of Cambridge, Cambridge, UK
[2]RAND Europe, Westbrook Centre, Cambridge, UK
[3]Academic Unit of Health Economics, Leeds Institute of Health Sciences, University of Leeds, Leeds, UK
[4]MRC Epidemiology Unit, University of Cambridge, Institute of Metabolic Science, Cambridge, UK
[5]Faculty of Health, Social Care and Education, Anglia Ruskin University, Cambridge, UK

**Correspondence to**
Dr Juliet A Usher-Smith;
jau20@medschl.cam.ac.uk

## ABSTRACT

**Objective** To synthesise data concerning the views of commissioners, managers and healthcare professionals towards the National Health Service (NHS) Health Check programme in general and the challenges faced when implementing it in practice.

**Design** A systematic review of surveys and interview studies with a descriptive analysis of quantitative data and thematic synthesis of qualitative data.

**Data sources** An electronic literature search of MEDLINE, Embase, Health Management Information Consortium, Cumulative Index of Nursing and Allied Health Literature, Global Health, PsycInfo, Web of Science, OpenGrey, the Cochrane Library, NHS Evidence, Google Scholar, Google, ClinicalTrials.gov and the International Standard Randomised Controlled Trial Number registry from 1 January 1996 to 9 November 2016 with no language restriction and manual screening of reference lists of all included papers.

**Inclusion criteria** Primary research reporting views of commissioners, managers or healthcare professionals on the NHS Health Check programme and its implementation in practice.

**Results** Of 18 524 citations, 15 articles met the inclusion criteria. There was evidence from both quantitative and qualitative studies that some commissioners and general practice (GP) healthcare professionals were enthusiastic about the programme, whereas others raised concerns around inequality of uptake, the evidence base and cost-effectiveness. In contrast, those working in pharmacies were all positive about programme benefits, citing opportunities for their business and staff. The main challenges to implementation were: difficulties with information technology and computer software, resistance to the programme from some GPs, the impact on workload and staffing, funding and training needs. Inadequate privacy was also a challenge in pharmacy and community settings, along with difficulty recruiting people eligible for Health Checks and poor public access to some venues.

**Conclusions** The success of the NHS Health Check Programme relies on engagement by those responsible for its commissioning, management and delivery. Recognising and addressing the challenges identified in this review, in particular the concerns of GPs, are important for the future of the programme.

### Strengths and limitations of this study

▶ This is the first study to systematically synthesise data concerning the views of commissioners, managers and healthcare professionals on the National Health Service Health Check programme.
▶ By including quantitative and qualitative data and studies not published in the mainstream medical literature, it provides a comprehensive overview.
▶ However, the included studies were at risk of selection bias with recruitment consistently reported to have been difficult, and all included only small sample sizes.
▶ Participants may also have responded in ways that reflected best practice or views they felt they ought to hold rather than their true views.

## INTRODUCTION

Despite improvements in clinical care and reductions in risk factors such as smoking, cardiovascular disease (CVD) remains the leading cause of years of life lost in the UK,[1] with nearly 400 people dying each day from CVD across England and Wales.[2] To help reduce this burden of disease, the National Institute for Health and Care Excellence[3 4] and WHO[5] recommend incorporating primary prevention initiatives. To address this, in 2009, Public Health England (PHE) introduced the National Health Service (NHS) Health Check programme in England. The aim of the programme is to offer to all those between 40 and 74 years of age, with no pre-existing CVD, type 2 diabetes or dementia, an assessment of their risk of developing CVD and diabetes and advice about risk management, including medication, lifestyle advice and referral services.

The NHS Health Checks are held in general practice (GP) surgeries, pharmacies and community settings and are delivered by

GPs, practice nurses, healthcare assistants (HCAs), pharmacists and/or pharmacy assistants. Although it has been a mandated public health service since 2013 with clear guidelines on the required elements,[6] there is flexibility in how local areas choose to commission the programme with GP surgeries and pharmacies choosing whether to deliver NHS Health Checks. The programme itself has also remained controversial, and its effectiveness has been challenged by both researchers and clinicians.[7–9] The result has been variability in approach to implementation and delivery across the country[10] and varying levels of engagement among healthcare professionals.

As with all individual-level interventions, the impact of the NHS Health Check programme depends on those delivering it. This review synthesises studies describing the views of commissioners, managers and healthcare professionals towards the NHS Health Check programme and in doing so explores some of the reasons behind this variation and the challenges faced when implementing the programme.

## METHODS

We performed a systematic literature review following a study protocol (available at https://osf.io/amb4z) that followed the Preferred Reporting Items for Systematic Reviews and Meta-Analyses guidelines.[11]

### Search strategy

Published studies were identified from the results of an existing literature review conducted by PHE covering the period from 1 January 1996 to 9 November 2016.[12] This was supplemented by a search in Web of Science and OpenGrey over the same time period. We undertook hand searches of the reference lists of all included publications and performed additional online searches for further publications by named authors identified in the search. Searches completed by PHE included the following sources: MEDLINE, PubMed, Embase, Health Management Information Consortium, Cumulative Index of Nursing and Allied Health Literature, Global Health, PsycInfo, the Cochrane Library, NHS Evidence, Google Scholar, Google, ClinicalTrials.gov and the International Standard Randomised Controlled Trial Number registry. Full details of all the search strategies are shown in online supplementary appendix 1. No language restrictions were applied.

### Study eligibility criteria

Study selection was a two-part process. Initially, studies were screened by title and abstract for potential relevance to the NHS Health Checks. We excluded commentaries, editorials and opinion papers. In the second stage, we identified studies reporting the views and experiences of healthcare professionals on NHS Health Checks. Two researchers (JUS and AM) read the full-text of all the potentially relevant studies. Studies for which it was unclear whether or not these inclusion criteria were met

were discussed at consensus meetings with the wider research team.

### Data extraction, quality assessment and synthesis

Data extraction was completed independently by two researchers (JU-S+AM/CLS/KM for the quantitative data and JU-S+EH/CMa/KM for the qualitative data). Data extracted included study design, time period, recruitment method, participants and analytical method. Studies were also assessed for quality using the Critical Appraisal Skills Programme (CASP) checklist[13] for qualitative studies or a combined CASP checklist for cohort or randomised controlled trials for the quantitative studies. No studies were excluded on the basis of quality alone.

We synthesised the qualitative data using thematic synthesis approaches which have been described in detail elsewhere.[14] Briefly, after initial reading and re-reading of the papers, we first coded all findings under the headings of 'results' and findings' within the primary studies. We then organised these codes into descriptive, and subsequently analytical, themes. The initial coding was completed by two researchers (JU-S and EH/CMa). Each researcher had experience of conducting and analysing qualitative data and brought their own professional background (academic GP, public services, health systems and innovation) to the interpretation of the findings. Consensus meetings were held with the wider research team, which included researchers with both clinical and non-clinical backgrounds and those with relevant topic expertise, to discuss the emerging codes and develop descriptive and analytical themes. To allow for appreciation of the data reviewed in these studies, illustrative quotations have been included alongside the analytical themes presented.

For the quantitative data, we extracted all the findings from the studies and synthesised those descriptively, grouping similar aspects together.

## RESULTS

The initial literature search generated 18 524 titles and abstracts. One hundred and seventy-eight papers were potentially relevant to NHS Health Checks. These were reviewed at full-text level (figure 1). Of those, 164 were excluded. Reasons were that they did not include any relevant data for this research question, were duplicates or commentaries, or did not describe NHS Health Checks. Through citation searching, one additional article was identified. This review is, therefore, based on 15 articles.[15–29] The characteristics of these are shown in table 1, and the detailed quality assessment is shown in tables 2A and 2B.

The studies used a range of designs. One included quantitative results from surveys,[19] two quantitative results from surveys alongside free-text question responses,[15 18 22] one free-text responses from a survey[29] and 10 findings from semistructured

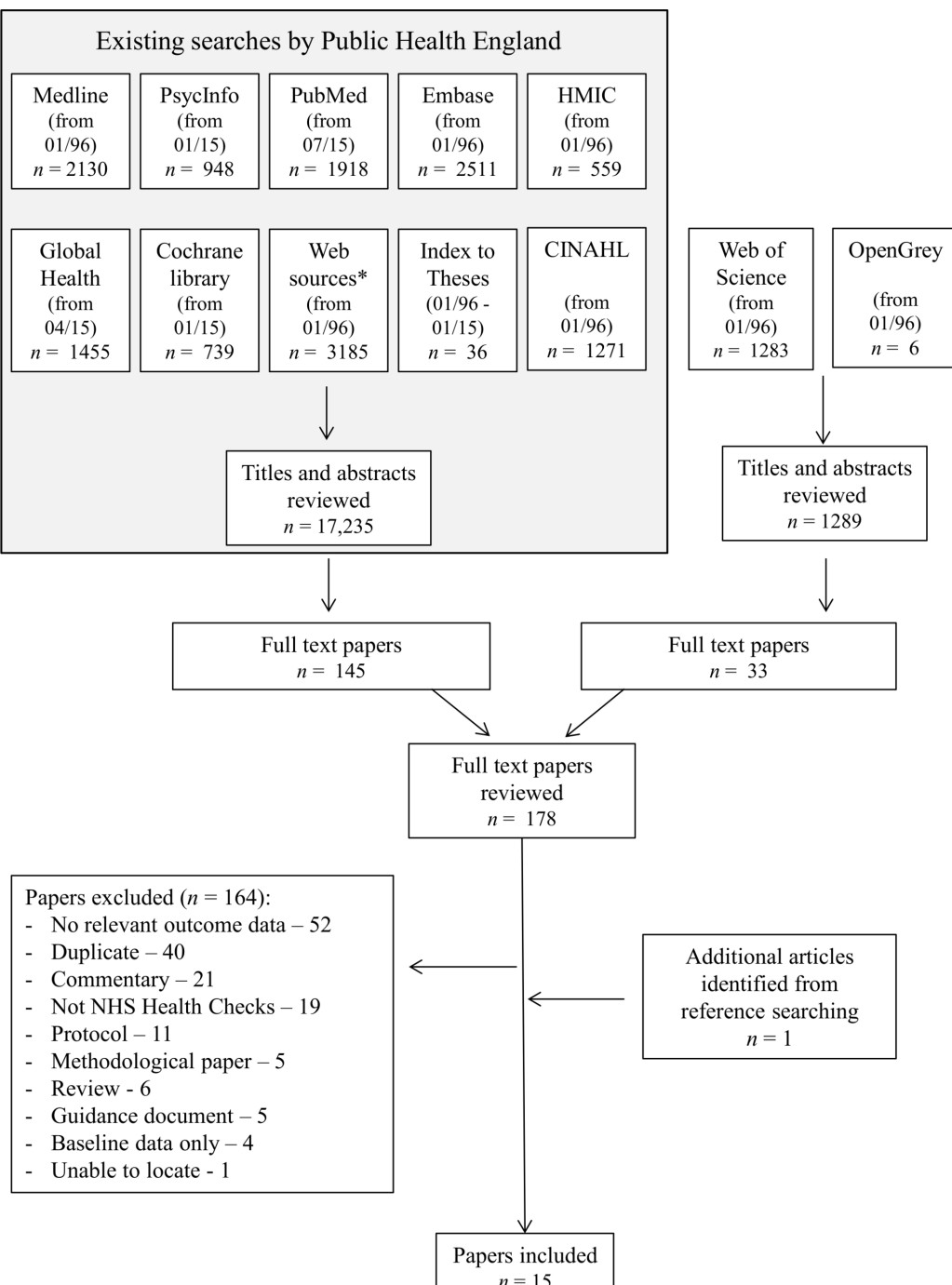

**Figure 1** PRISMA flow diagram. CINAHL, Cumulative Index of Nursing and Allied Health Literature; HMIC, Health Management Information Consortium; NHS, National Health Service; PRISMA, Preferred Reporting Items for Systematic Reviews and Meta-Analyses.

interviews.[16 17 20 21 23–28] The majority (n=10) reported the views of healthcare professionals working within GP. Four included pharmacists,[17 19 21 23] four those delivering NHS Health Checks within community settings[23 25 26 29] and two commissioners.[21 23] Most collected data within the first two years of the programme (2009–2011). Sample sizes ranged between 25 and 442 for the survey studies and between 4 and 58 for the interview studies. All of the qualitative studies were considered to be of medium or high quality, and the three quantitative studies were all of medium quality. Response rates for the two survey studies that reported them were 24% for GPs,[18] 76% for practice managers[18] and 34% for pharmacists.[19]

### Overall views of the NHS Health Check programme
#### Commissioners
Only one study reported the views of commissioners on the programmes as a whole.[23] Across the 14 commissioners interviewed, their enthusiasm for NHS Health

**Table 1** Features of studies

| Author, year | Type of report | Study period | Location of study | Setting of NHS Health Check | n | Data collection method | Method of recruitment to study | Participant characteristics | Method of analysis |
|---|---|---|---|---|---|---|---|---|---|
| Baker et al, 2015[15] | Journal article | Not given | South West England | 30.1% of total practices delivering NHS Health Checks | 25 | Surveys including quantitative and qualitative questions | Identified randomly via the County Medical List to ensure geographical spread | 2 GPs, 14 practice managers, 6 practice nurses, 2 HCAs and 1 administrator | Descriptive statistics Thematic analysis |
| Crabtree et al, 2010[17] | Conference abstract | 2009 | Not given | 32 (of 35) pharmacies in the area delivering NHS Health Checks | 32 | Semistructured telephone interviews | All 35 pharmacies delivering the service were contacted | 15 pharmacists, 13 support staff and 4 pre-registration pharmacists | Thematic analysis |
| NHS Greenwich, 2011[29] | Report | 2011 | Greenwich | Community | 11 | Open-ended questionnaire | All (12) clinicians delivering community outreach services providing NHS Health Checks were invited | HCAs, nurses, pharmacists and health trainers | Not described |
| Ismail and Kelly, 2015[16] | Journal article | 2010 | Yorkshire | 25 GPs | 58 | Semistructured interviews | Letters of invitation or flyers to 41 GPs targeted to reflect diversity in terms of performance | HCAs, GPs, practice managers, practice nurses and other support staff | Framework analysis |
| Krska et al, 2016[18] | Journal article | 2011 | Sefton, an area of North West England | 33 (of 55) GPs | 83 (76% of practice managers and 24% of GPs) | Postal survey with free-text responses | Personally addressed letters of invitation with a covering letter to all practice managers and GPs at 55 practices | 40 practice managers and 43 GPs | Categorisation of responses |
| Loo et al, 2011[19] | Conference abstract | 2009 | Not given | Pharmacies | 442 (34%) | Postal questionnaire | Questionnaire posted to all pharmacies in the area | All pharmacists 59% men; 89.1% full-time; 53.4% worked for large multiple pharmacies | Descriptive statistics |

Continued

**Table 1** Continued

| Author, year | Type of report | Study period | Location of study | Setting of NHS Health Check | Data collection method | n | Method of recruitment to study | Participant characteristics | Method of analysis |
|---|---|---|---|---|---|---|---|---|---|
| McDermott et al, 2016[20] | Journal article | 2013–2015 | 2 London boroughs | 17 GPs | Semistructured interviews | 24 | Recruited from within a trial of an enhanced invitation method | 52% practice managers, 9% HCAs, 30% administrators, 9% public health leads | Framework analysis |
| McNaughton et al, 2011[21] | Journal article | Not given | Tees Valley | 8 pharmacies | Semistructured interviews | 20 | Postal invitation | 10 primary care trust members, 8 pharmacists, 2 representatives from Local Pharmaceutical Committee | Thematic analysis |
| Nicholas et al, 2013[22] | Journal article | 2011 | 2 London boroughs | 70 (of 96) GPs | Survey with free-text responses | 65 | Invitations to all 96 GPs | 25 practice managers, 8 GPs, 16 practice nurses, 2 HCAs, 3 administrators and 14 not specified | Descriptive statistics Content analysis |
| Oswald et al, 2010[28] | Evaluation report | 2009–2010 | Teesside | 13 GPs | Semistructured interviews | 25 | Letter of invitation to practice managers | 8 practice managers, 14 practice nurses, 1 GP, 1 HCA, 1 pharmacist | Thematic analysis |
| Research Works, 2013[23] | Research report | 2013 | Not given | All settings | Semistructured interviews | 26 | Contacts provided by commissioners with snowballing recruitment | 14 commissioners, 12 GPs, practice managers, healthcare assistant, nurse practitioner, physical activity development officer, health bus workers and a community pharmacist | Not described |

Continued

**Table 1** Continued

| Author, year | Type of report | Study period | Location of study | Setting of NHS Health Check | Data collection method | n | Method of recruitment to study | Participant characteristics | Method of analysis |
|---|---|---|---|---|---|---|---|---|---|
| Riley et al, 2015[25] | Journal article | 2013 | Bristol inner city | Community settings | Semistructured interviews | 4 | Participants were recruited via their involvement with community outreach events | 1 practice nurse, 1 HCA, 1 engagement worker and 1 health trainer | Thematic analysis |
| Riley et al, 2016[24] | Journal article | 2013–2014 | Bristol | 11 GPs | Semistructured interviews | 15 | 18 were invited with purposive sampling | 5 GPs, 5 practice nurses, 3 HCAs, 2 pharmacists | Thematic analysis |
| Shaw et al, 2015[26] | Journal article | 2010–2011 | Birmingham and Black Country | GPs and community | Semistructured interviews | 31 | Recruited through lead clinicians | 9 GPs, 6 practice managers, 4 practice nurses, 6 HCAs, 1 alternative provider director, 1 call centre manager, 2 call centre operatives and 2 alternative provider registered practice nurses | Thematic analysis |
| Shaw et al, 2016[27] | Journal article | Not given | Birmingham | GPs | Semistructured interviews | 9 | Recruitment undertaken by local NHS trust. No further details were provided | All GPs | Thematic analysis |

GP, general practice; HCA, healthcare assistant; NHS, National Health Service.

**Table 2A** Quality assessment of studies including surveys

| Author, year | Study addressed a clearly focused issue | Use of an appropriate method/randomisation (for RCTs) | Recruitment/ comparability of study groups at baseline | Blinding (for RCTs) | Exposure measurement | Outcome measurement | Comparability of study groups during study (for RCTs) | Follow-up (for longitudinal studies) | Confounding factors (for non-RCTs) | Applicability to England | Overall |
|---|---|---|---|---|---|---|---|---|---|---|---|
| Baker et al, 2015[15] | *** | † | ** | n/a | n/a | ** | n/a | n/a | * | * | Medium |
| Krska et al, 2016[18] | *** | *** | ** | n/a | n/a | *** | n/a | n/a | * | ** | Medium |
| Loo et al, 2011[19] | *** | *** | ** | n/a | n/a | n/a | n/a | n/a | * | ** | Medium |

***High.
**Medium.
*Low.
RCT, randomised controlled trial.

**Table 2B** Quality assessment of studies including qualitative data

| Author, year | Study addressed a clearly focused issue | Appropriateness of qualitative method | Design | Recruitment | Consideration of relationship between research and participants | Ethical issues | Rigour of data analysis | Clarity of statement of findings | Overall |
|---|---|---|---|---|---|---|---|---|---|
| Baker et al, 2015[15] | *** | *** | *** | * | * | ** | ** | *** | Medium |
| Crabtree et al, 2010[17] | *** | *** | * | * | * | ** | * | ** | Medium |
| NHS Greenwich, 2011[29] | *** | *** | *** | ** | * | * | * | ** | Medium |
| Ismail and Kelly, 2015[16] | *** | *** | *** | ** | ** | *** | *** | *** | High |
| Krska et al, 2016[18] | *** | *** | ** | *** | n/a | *** | ** | ** | Medium |
| McDermott et al, 2016[20] | *** | *** | *** | * | * | *** | ** | ** | Medium |
| McNaughton et al, 2011[21] | *** | *** | *** | *** | * | *** | *** | *** | High |
| Nicholas et al, 2013[22] | *** | *** | *** | *** | n/a | *** | *** | *** | High |
| Oswald et al, 2010[28] | *** | *** | *** | ** | * | *** | * | ** | Medium |
| Research Works, 2013[23] | *** | *** | *** | ** | * | *** | * | ** | Medium |
| Riley et al, 2016[24] | *** | *** | *** | ** | ** | *** | *** | *** | High |
| Riley et al, 2015[25] | *** | *** | *** | ** | ** | *** | *** | *** | High |
| Shaw et al, 2015[26] | *** | *** | *** | ** | *** | *** | *** | *** | High |
| Shaw et al, 2016[27] | *** | *** | *** | ** | * | *** | *** | *** | High |

***High.
**Medium.
*Low.

Checks varied: whereas many approached the programme positively, others described lower levels of engagement.

> It's very difficult to provide reassurance when on a personal level you're not sure if you've 100% bought into the programme either.—Commissioner[23]

### GP healthcare professionals

Two studies reported quantitative results from surveys with GP healthcare professionals. In one, a survey of 43 GPs from 31 practices,[18] 51% (n=22) viewed the programme as important, 54% (n=24) as beneficial to their patients and 5% (n=2) considered the NHS Health Check programme to be a waste of time and resources. In the same study, 36 out of 81 GPs and practice managers (44%) felt the high-risk patient identification was beneficial to the practice. In a second survey of 25 healthcare professionals, 72% (n=18) perceived that NHS Health Checks were useful in early detection and gave time to discuss patient health and lifestyles.[15]

Of the 10 interview studies, in general, participants expressed the view that NHS Health Checks were beneficial in the early detection and prevention of disease.[16 20 23 27]

> It's a good way to try and prevent illness and long term or serious conditions developing in the future.—Practice manager[20]

> I think it's a very good idea. We have a very high proportion of our patients who suffer with diabetes, almost 10% of our patients are diabetic so I thought this was an excellent opportunity to screen those earlier and pick them up. GP[27]

There were, however, a number of concerns raised about the programme. In particular, some GPs described how they felt the programme attracted the 'worried well' and that the patients who would benefit the most were the ones who were least likely to attend.[15 16 22–24]

> If you send out an invite to a large number of people then the people who present themselves (laughs) er might well fit into that worried well category, um won't necessarily be um the HGV driver who works long hours and smokes a lot.—GP[24]

Many also described doubts about the long-term benefits and the costs of implementation, including staff resources and lack of evidence for the effectiveness.[15 16 18 20 24]

> I don't think there is an awful lot of value. I think you'll pick up a few people a little bit earlier. Now whether that's worth the cost, obviously it's great for those individual patients, whether that's worth the cost of running a programme like this. I'd be amazed if it was.—Nurse[24]

> I think really this is mass screening and there's not a great deal of proof behind it…. Not entirely convinced with being told we have to offer a check to everyone.—GP[16]

Linked to this, participants in several of the studies described the challenges to achieving behaviour change, and the difficulties they had getting people to make long-standing changes to their lifestyle following the health checks.[15 16 26 27]

> Even if you access them, even if you find out that they're a really high risk score then getting these people to take on board you know the lifestyle changes, changes to their diet, exercising more. It's very difficult to get them to take those changes on.—Nurse[16]

Managing high-risk levels of alcohol consumption was felt to be especially challenging for some GPs and staff, particularly among patients in certain religious groups in which alcohol consumption can be stigmatised.[27] A lack of resources and lack of, or inconsistency of, well-funded support services in the wider community also contributed to this.[15 16 23 27]

> We used to have things called exercise referral and we refer people to free gym sessions and send them to Slimming World and they'd get Slimming World sessions. We had really good responses and really good uptake for that, but that's all gone now.—Nurse[16]

### Pharmacists

Three studies described the views of pharmacists.[17 19 21] Two of these are conference abstracts in which pharmacists and those involved in the delivery of NHS Health Checks in pharmacies had been interviewed.[17 19] The third sent out a postal questionnaire to pharmacists, reporting a 34% response rate.[21] In contrast to the studies with healthcare professionals from GP, very few participants from pharmacies discussed the benefits or otherwise of the NHS Health Checks to patients. Instead, the focus was on the benefits of delivering NHS Health Checks in pharmacies, with all feeling it offered immense job satisfaction, promoted the image of the pharmacy and provided a good opportunity for staff development.[17 19 21]

> I wanted to do this regardless…. if I'm in a position where I can give somebody information that will then enable them to change their behaviour and live a healthier life that's a satisfying thing to do.—Pharmacist[17]

> For being the place to come in your local area for your health concerns, I think all round, for both the staff personally and for the company's goal, I think it's a positive thing.—Pharmacist[17]

### Those delivering NHS Health Checks in community settings

No studies reported the views towards the programme as a whole from those involved in delivering NHS Health Checks in community settings.

### Challenges to implementation

One study reported challenges to implementation across all settings reported by commissioners.[23] In that

**Table 3** Challenges to implementation of NHS Health Checks reported across the settings

| Challenge to implementation | GPs | Pharmacies | Community settings |
|---|---|---|---|
| Difficulties with IT and computer software | ✓[16 18 22 23] | ✓[21 23] | ✓[23 25 29] |
| Impact on workload/staffing | ✓[15 16 18 22] | ✓[17 19 21] | |
| Funding | ✓[16 18] | ✓[19 21] | |
| Training needs | ✓[15 16 22 27] | ✓[19 21] | |
| Resistance from GPs | ✓[23] | ✓[23] | ✓[23] |
| Inadequate privacy | | ✓[19 21 23] | ✓[25 29] |
| Difficulty recruiting participants | | ✓[21 23] | |
| Poor access to some venues | | | ✓[29] |

GP, general practice; IT, information technology; NHS, National Health Service.

study, the greatest challenges were: engaging with GPs, both to encourage them to deliver NHS Health Checks within their practice and to facilitate delivery by non-GP providers; difficulties with data management in the absence of standard Read Codes when NHS Health Checks were first introduced and the lack of clear national guidelines around data handling and ensuring consistency of provision across GPs, particularly with the lack of a formal quality assurance or monitoring system at the time.

> The massive thing is the sheer variability in delivery. You get some star performers and some people that just won't engage with it.—Commissioner[23]

### General practice

Seven studies described the challenges GP healthcare professionals had experienced when implementing the NHS Health Checks within their practice. The main challenges are summarised in table 3. Difficulties with information technology (IT) and computer software were mentioned in over half of the studies, particularly related to the call and recall system when the programme was introduced.[16 18 22 23] 39% of practice managers in one study reported difficulties with the clinical system, software or errors in the existing data.[18] Impact on workload was also cited as a challenge for some. In a survey of 25 healthcare professionals, approximately 40% indicated that there had been issues with staffing levels since starting to deliver NHS Health Checks, with some attributing these issues to the extra workload created by NHS Health Checks.[15]

> NHS Health Check generates a huge workload for our staff in addition to what we do, a roughly 20 per cent additional workload.—Nurse[15]

In other studies, practice managers and GPs also generally agreed that the programme's impact on workload had knock-on effects on other services,[18] with the financial reimbursement considered not sufficient to justify the work[18 23 27] or influencing their implementation.[27]

> In order to get good payments we had to reach 50% target within three months… it was important for us to get the targets very very quickly.—GP[27]

Concerns about remuneration were also reported by commissioners who claimed that NHS Health Checks were less of a priority as they are not part of the Quality and Outcomes Framework for which GPs get paid.[23]

> GPs have a very 'small business' mentality, and if the Health Care Assistant is off doing a Health Check and can't be doing something else for them, they get very jittery about that.—Commissioner[23]

Inadequate training was the final theme and was discussed in many of the studies.[15 16 22 27] These include a survey of 25 healthcare professionals in which 44% (n=11) indicated that they required further training.[15] A survey of staff at 65 GPs in two inner London boroughs showed that staff at 62% (n=40) and 65% (n=42) of practices had attended training in delivering lifestyle advice or risk information, but only 43% (n=28) of practices reported that staff had attended training in measurement methods; at 23% (n=15) of practices, no specific training was reported, and 28% (n=18) considered that additional training would have been beneficial.[22] In free-text responses, 24% (n=5/21) of healthcare professionals suggested that improvements to staff training and capacity were required.[22]

> [Training} would be good. As I say, we just learnt from our healthcare assistant what to do; basically it was like kind of on the job training… It would be nice to understand it in depth more, wouldn't it?—HCA[16]

### Pharmacies

Three studies, two of which are conference abstracts, reported the challenges faced by those involved in commissioning or delivering NHS Health Checks within pharmacies. In a survey of 442 community pharmacists,[19] the three most important perceived barriers to implementation were lack of time, lack of staff and lack of reimbursement (all reported by over 55% of respondents). Lack of time and staff were also referred to in qualitative interviews with pharmacy staff. In particular, they described how, owing to other commitments, most pharmacists did not have the capacity to perform the initial assessments as part of the NHS Health Checks. Instead, these were carried out by pharmacy assistants, who in turn needed more substantial training than was initially offered[17 19 21]

The people they have working for them… haven't got the background in care knowledge or expertise. It wasn't like a GP surgery where you have Healthcare Assistants and Practice Nurses who on a day to day basis take blood pressures, take pulses, take blood and give advice on health.—PCT staff member[21]

Difficulties with funding were also discussed by commissioners who had had to develop different agreements from those with GP practices as pharmacies pay value added tax on all services they deliver and had had to allocate additional funds for unexpected costs, such as having to vaccinate pharmacy staff to allow them to handle blood and bodily fluids.[21]

Other challenges (table 3) identified by pharmacists and commissioners included: lack of private space for consultations (25%, n=111/442)[19]; difficulties with IT, particularly the need for a sufficiently secure internet connection to allow them to transfer patient identifiable data and difficulty recruiting participants as the eligible population was largely dictated by footfall within the pharmacy.[19 21 23] Some pharmacies that were very close to GP practices delivering the NHS Health Check also experienced competition between settings.

Actually there's another problem, capturing the people. Everyone is out to capture them… it's very hard if you see someone coming in and say, 'Oh! You could be a candidate', and they say, 'The surgery has approached me and I'm going there.—Pharmacy representative[21]

### Community settings
Three studies reported the views of those involved in delivering NHS Health Checks in community settings.[23 25 29] In contrast to some of the views expressed by HCAs working in GP, in a small study of 10 HCAs delivering community-based NHS Health Checks most felt there were enough staff and felt they had adequate support.[29] Workers on a Health Bus also found delivering NHS Health Checks to be fulfilling, enjoyable and overall a positive experience.[23] The main challenges identified (table 3) were poor access to some venues, inadequate privacy, problems with some of the equipment and connection to the internet and resistance from GPs to accept referrals from third-party providers.

I don't think you come across very professional when you're sitting in a kitchen and all huddled round and all on top of each other. And it's not very nice for the patients, because…quite personal information.—Nurse[25]

Because we were all in the same room it was easy to listen to what was happening next door.—HCA[29]

## DISCUSSION
### Key findings
While there was evidence that some commissioners, managers and healthcare professionals working in GP could see the benefit of the NHS Health Check programme for patients, in the largest survey of GPs, only half viewed the programme as important and beneficial to their patients. A range of views was also seen in qualitative studies where some were enthusiastic, whereas others raised concerns around inequality of uptake, the evidence behind the programme and the cost-effectiveness. In contrast, those working in pharmacies were all positive about the programme, citing opportunities for their business and staff as reasons.

A number of challenges to implementation were identified. Difficulties with IT and computer software and resistance to the programme from GPs were described across all settings. The impact on workload and staffing, funding and training needs were also challenges in GP and pharmacy settings, whereas inadequate privacy was common to both pharmacies and community settings. Some pharmacies also experienced difficulty recruiting people for NHS Health Checks, and poor access to some venues was reported in community settings.

### Strengths and limitations
The strengths of this review include the comprehensive electronic search across multiple databases, the inclusion of reports not published within the mainstream medical literature and the synthesis of both quantitative and qualitative data. However, all the studies included only small sample sizes and so may not be generalisable beyond the study context. In addition, recruiting GPs was consistently reported to have been difficult, especially from practices performing fewer NHS Health Checks, and the pharmacists who took part were all from pharmacies already involved in delivering NHS Health Checks; these studies are therefore at a particular risk of selection bias. Although the studies included a range of professionals from different settings, the views reported may, therefore, reflect the opinions of those who are particularly enthusiastic or negative or have strong views about the NHS Health Check programme. The findings are also constrained by the questions addressed by the original researchers. Second, across all the studies, it is possible that participants responded in ways that reflected best practice or the views they felt they ought to hold, and so the findings may not reflect their true personal views. We also did not have access to the original data and so were only able to synthesise the findings considered by the authors of the original studies as worthy of report. Finally, all but two studies were conducted prior to 2013 and so are more representative of the initial phase of the programme and may not reflect changes since then.

### Comparison with existing literature
While we only included studies specific to the NHS Health Check programme, the main challenges to implementation identified in this study are consistent with those reported for prevention and health promotion in general. A multinational study across 11 European countries which included over 2000 GPs found that, although GPs

believed prevention and health promotion was important, the workload, lack of time and need for funding limited their engagement.[30] Issues around workload and lack of time were also the two main barriers in a survey of GP views of their role in cancer prevention in the UK[31] and, along with lack of funding, were reported in a questionnaire survey of GP healthcare professional's views on advising patients about physical activity[32] and a qualitative study of lifestyle counselling in Ireland.[33] The concerns expressed by some healthcare professionals in this study about the difficulties changing patients' behaviours are also commonly reported in the literature[32 34–36]: in one survey, 40.3% (n=112/278) of GPs agreed that patients' behaviours are established and difficult to change.[31]

### Implications for clinicians, policymakers and future research

Given their central role in the success of the programme, the finding that a number of commissioners, GPs and other GP staff had doubts about the evidence behind the programme has important implications for future delivery of NHS Health Checks. Lack of belief about proven effectiveness has been identified as one of the main barriers to offering health promotion activities within routine care among Dutch GPs and nurses[36] and evidence of effectiveness as one of the main incentives for GPs in a multinational study.[30] A survey of Australian GPs' views on clinical guidelines also cited an evidence base as the most important factor in their deciding whether to follow the recommendations of a guideline.[37] In the eight years since the programme was introduced, there has been a growing evidence base around the NHS Health Checks. In contrast to the views held by many of the healthcare professionals in these studies, evidence suggests, for example, that, in part owing to targeted approaches, more people in the most deprived quintile compared with the least deprived quintile have had NHS Health Checks,[38–40] and there has been a consistent 3% to 4% increase in statin prescribing among attendees of the NHS Health Check compared with matched non-attendees.[41–43] Ensuring that programmes are effective and producing up-to-date, concise, summaries of the evidence and estimated benefits for different patient groups in an easily accessible format should therefore be a priority for those supporting delivery of the NHS Health Check programme and other similar prevention programmes. Piloting future programmes to provide such evidence before rolling them out nationally and including a phased roll out with in-built evaluation may also help address some of these concerns, particularly among GPs whose engagement is key to delivery of the programme in all settings.

Anticipating and addressing training needs and difficulties with IT and computer software early may also increase engagement. Indeed, in 2013, since the majority of these studies were published, PHE introduced standard Read Codes to facilitate data entry, updated software for identifying those eligible and provided additional online training modules for healthcare professionals.

Overcoming some of the other challenges identified, such as funding and increased workload, are more difficult given the context of the current financial crisis within the NHS and reports of primary care services being stretched beyond safe limits by the needs of those with existing morbidity.[44] However, this review suggests that there may be greater enthusiasm among pharmacists than GPs for delivering NHS Health Checks. Capitalising on this may be an effective way to reduce pressure on GPs while at the time empowering pharmacists to take on a wider role within healthcare.[45]

**Acknowledgements** We thank our patient and public representatives Kathryn Lawrence and Chris Robertson for providing helpful comments on the findings and the NHS Health Checks Expert Scientific and Clinical Advisory Panel working group for providing us with the initial literature search conducted by Public Health England. We would also like to thank Anna Knack, Research Assistant at RAND Europe, for her excellent research support, and Emma Pitchforth for her helpful comments on our analysis. A summary of the findings reported in this manuscript has been published online by Public Health England (available at http://www.healthcheck.nhs.uk/commissioners_and_providers/evidence/) and RAND (http://www.rand.org/content/dam/rand/pubs/external_publications/EP60000/EP67129/RAND_EP67129.pdf). Permission from both has been obtained to publish the results in this journal.

**Contributors** KM synthesised and interpreted the findings and wrote the first draft of the manuscript. EH screened articles for inclusion, extracted and synthesised the qualitative data, interpreted the findings and critically revised the manuscript. CMa extracted and synthesised the qualitative data and critically revised the manuscript. AM screened articles for inclusion, interpreted the findings and critically revised the manuscript. CS, CMe, FW, SG and JM developed the protocol, interpreted the findings and critically revised the manuscript. JU-S developed the protocol, screened articles for inclusion, extracted and synthesised the quantitative and qualitative data, interpreted the findings and wrote the first draft of the manuscript.

**Funding** This work was funded by a grant from Public Health England. JU-S and KM are funded by a Cancer Research UK/BUPA Foundation Cancer Prevention Fellowship (C55650/A21464) and FW by an NIHR Clinician Scientist award. The views expressed in this publication are those of the authors and not necessarily those of the NHS, the NIHR or the Department of Health. All researchers were independent of the funding body, and the funder had no role in data collection, analysis and interpretation of data; writing of the report; or decision to submit the article for publication.

**Provenance and peer review** Not commissioned; externally peer reviewed.

**Data sharing statement** All data are available from the reports or authors of the primary research. No additional data are available.

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
