## [Reviewer comments · BMJ Open]

ARTICLE DETAILS

TITLE (PROVISIONAL)	Views of commissioners, managers and healthcare professionals on the NHS Health Check programme: a systematic review
AUTHORS	Mills, Katie; Harte, Emma; Martin, Adam; MacLure, Calum; Griffin, Simon; Mant, Jonathan; Meads, Catherine; Saunders, Catherine; Walter, Fiona; Usher-Smith, Juliet

VERSION 1 – REVIEW

REVIEWER	Dr Richard Cooke, Senior Lecturer in Health Psychology Aston University, UK
REVIEW RETURNED	15-Aug-2017

GENERAL COMMENTS	Really good paper. Three minor points. First, I recommend the authors publish their protocol on the Open Science Framework, and, if doing systematic reviews in the future, submit a protocol to PROSPERO. Second, top two lines of page 5 mention a previously published report for Public Health England. It would be good to include a reference for this report. Finally, when describing the number of studies in a review, such as on line 42 on page 6, that do something 'The majority (n=10)...' it is conventional to replace n with k. This is because n refers to the number of individuals whereas k refers to the number of papers.
---

REVIEWER	Anne-Louise Bjerregaard Aarhus University, Department of Public Health, Section of General Medical Practice, Aarhus, Denmark
REVIEW RETURNED	25-Aug-2017

GENERAL COMMENTS	The authors review the current literature that evaluates views on the NHS Health Check programme. The manuscript is well-written and will provide a brief overview of the views of commissioners, managers and health care professionals on the NHS Health Check programme. Knowledge on the potential challenges in maintenance and running of the programme is relevant in order to improve quality and hence the future results of the NHS programme. Strengths of the study is the utilization of the 'prisma' checklist and guidelines for evaluating qualitative studies in the synthesis of the evidence. The relevance is high, although there is a lack of newer studies, and thus, the views given in the review might be more representative for the initial phase of the NHS Health Check programme.
--

	The review only synthesizes data from the NHS Health Check Programme (rather than Health Check programmes in general) and as such the 'views' are limited to the specific setting in the NHS (although they can to some extent be translated to other European health care systems). I have only one minor comment: 1) The authors include two studies, based on conference abstracts. I wonder why these abstracts (from 2010/2011) are not published as peer-reviewed papers yet? The authors could perhaps include a sentence in the results section (page 9, line 6-9) noting that this is based on conference abstracts only.
--	---

VERSION 1 – AUTHOR RESPONSE

Reviewer: 1

Reviewer Name: Dr Richard Cooke, Senior Lecturer in Health Psychology Institution and Country: Aston University, UK Competing Interests: None declared

Comment: Really good paper. Three minor points. First, I recommend the authors publish their protocol on the Open Science Framework, and, if doing systematic reviews in the future, submit a protocol to PROSPERO. Second, top two lines of page 5 mention a previously published report for Public Health England. It would be good to include a reference for this report. Finally, when describing the number of studies in a review, such as on line 42 on page 6, that do something 'The majority (n=10)...' it is conventional to replace n with k. This is because n refers to the number of individuals whereas k refers to the number of papers.

Response: We are pleased that you enjoyed our paper. As you suggested, we have published our protocol on the Open Science Framework and added the following text to the manuscript to reflect that:

“We performed a systematic literature review following a study protocol (available at osf.io/amb4z) that followed the PRISMA guidelines.”

We have also included a reference to the website where the existing literature review mentioned at the top of page 5 can be found (http://www.healthcheck.nhs.uk/commissioners_and_providers/evidence/literature_review/).

With regards to your suggestion to replace n with k when referring to the number of papers, it is not used within the Cochrane handbook for systematic review or mentioned within the PRISMA statement. We have therefore left these as n within the manuscript.

If the editorial team would prefer us to replace these with k we would however be happy to do that.

Reviewer: 2

Reviewer Name: Anne-Louise Bjerregaard

Institution and Country: Aarhus University, Department of Public Health, Section of General Medical Practice, Aarhus, Denmark
Competing Interests: None Declared

Review comments: "Views of commissioners, managers and healthcare professionals on the NHS Health Check programme: a systematic review"

The authors review the current literature that evaluates views on the NHS Health Check programme. The manuscript is well-written and will provide a brief overview of the views of commissioners, managers and health care professionals on the NHS Health Check programme. Knowledge on the potential challenges in maintenance and running of the programme is relevant in order to improve quality and hence the future results of the NHS programme.

Response: We are pleased that you felt our manuscript was well-written and relevant.

Comment: Strengths of the study is the utilization of the 'prisma' checklist and guidelines for evaluating qualitative studies in the synthesis of the evidence. The relevance is high, although there is a lack of newer studies, and thus, the views given in the review might be more representative for the initial phase of the NHS Health Check programme. The review only synthesizes data from the NHS Health Check Programme (rather than Health Check programmes in general) and as such the 'views' are limited to the specific setting in the NHS (although they can to some extent be translated to other European health care systems).

Response: We agree with both these points and so have added the following text into the "Strengths and limitations" and "Comparison with existing literature" sections of the manuscript:

"Finally, all but two studies were conducted prior to 2013 and so are more representative of the initial phase of the programme and may not reflect changes since then."

"While we only included studies specific to the NHS Health Check programme, the main challenges to implementation identified in this study are consistent with those reported for prevention and health promotion in general."

I have only one minor comment:

The authors include two studies, based on conference abstracts. I wonder why these abstracts (from 2010/2011) are not published as peer-reviewed papers yet? The authors could perhaps include a sentence in the results section (page 9, line 6-9) noting that this is based on conference abstracts only.

Response: We have double checked and neither of these articles have been published as peer-reviewed papers. As you suggest, we have included the following text in the results section to note this:

"Three studies described the views of pharmacists^{17,19,21}. Two of these are conference abstracts in which pharmacists and those involved in the delivery of NHS Health Checks in pharmacies had been interviewed^{17,19}."

"Three studies, two of which are conference abstracts, reported the challenges faced by those involved in commissioning or delivering NHS Health Checks within pharmacies."